# A Synthetic Cytokinin Improves Photosynthesis in Rice under Drought Stress by Modulating the Abundance of Proteins Related to Stomatal Conductance, Chlorophyll Contents, and Rubisco Activity

**DOI:** 10.3390/plants9091106

**Published:** 2020-08-27

**Authors:** Ranjit Singh Gujjar, Pennapa Banyen, Wannisa Chuekong, Phapawee Worakan, Sittiruk Roytrakul, Kanyaratt Supaibulwatana

**Affiliations:** 1Faculty of Science, Mahidol University, Rama VI Rd., Ratchathewi, Bangkok 10400, Thailand; ranjit.gujjar@icar.gov.in (R.S.G.); pennapa.by@gmail.com (P.B.); wannisa.chue@gmail.com (W.C.); phapawee.w@gmail.com (P.W.); 2Division of Crop Improvement, Indian Institute of Sugarcane Research, Lucknow 226002, India; 3National Center for Genetic Engineering and Biotechnology, National Science and Technology Development Agency, Pathum Thani 12120, Thailand; sittiruk@biotec.or.th

**Keywords:** photosynthesis, proteome, cytokinin, drought, rice

## Abstract

Drought susceptible rice cultivar PTT1 (Pathumthani1) was treated with drought (−72 kPa) and CPPU (N-2-(chloro-4-pyridyl)-N-phenyl urea) @ 5 mg/L at tillering and grain-filling stages. Plants were tested for the effect of synthetic cytokinin on the parameters influencing the process of photosynthesis. Exogenous spray of CPPU improved the stomatal conductance of rice leaves, which was severely reduced by drought. The abundance intensities of proteins, associated with the stomatal conductance (ZEP, NCED4, PYL9, PYL10, ABI5, SnRK4, Phot1, and Phot2), were also in agreement with the positive impact of CPPU on the stomatal conductance under drought stress. Among the photosynthetic pigments, Chl b contents were significantly reduced by drought stress, whereas CPPU treated plants retained the normal contents of Chl b under drought stress. Subsequently, we examined the abundance intensities of chlorophyll synthase and HCR proteins, implicated in the biosynthesis of chlorophyll pigments and the conversion of Chl b to Chl a, respectively. The results indicated a drought-mediated suppression of chlorophyll synthase. However, CPPU treated plants retained normal levels of chlorophyll synthase under drought stress. In addition, drought stress induced HCR proteins, which might be the cause for reduced Chl b contents in drought stressed plants. Further, CPPU treatment helped the plants sustain photosynthesis at a normal rate under drought stress, which was comparable with well-watered plants. The results were further confirmed by examining the abundance intensities of two key proteins, RAF1 and Rubisco activase, implicated in the assembly and activation of Rubisco, respectively. CPPU treatment reversed the drought mediated suppression of these proteins at both of the growth stages of rice under drought stress. Based on the results, it can be suggested that synthetic cytokinins help the plants sustain photosynthesis at a normal rate under drought stress by positively influencing the determinants of photosynthesis at a molecular level.

## 1. Introduction

Drought stress is prominent worldwide and impairs plant growth and development by affecting several biochemical and physiological processes. Plants counter drought stresses naturally by triggering a complex stress signaling cascade resulting in up or down-regulation of numerous regulatory and functional genes [1,2]. ABA (Abscisic acid) is a well-established stress hormone that triggers stress signaling in plants [3,4]. The ultimate aim of a plant during drought stress is to survive with minimal metabolic processes, which results in sluggish growth. Cytokinins play diverse roles in plant development, including cell growth and differentiation [5]. Recent findings suggest the role of cytokinins in mediating cellular responses to drought acclimation [6,7,8,9]. Cytokinins work antagonistically to ABA to regulate many developmental processes in plants during stress conditions [10,11]. High cytokinin concentration during osmotic stress counteracts leaf senescence by redistributing the remobilized nutrients [7,12,13], improves photosynthetic efficiency [14,15,16,17], interrupts drought-induced ABA responses [18,19], and eventually stops all those events that guide the plant to survive with minimal resources. There is ample evidence to suggest that cytokinins help in sustaining better plant growth under osmotic stress conditions, ultimately leading to improved yield [9,15,20,21,22].

Exogenous applications of synthetic cytokinins during osmotic stress improve MSI (membrane stability index), photosynthetic pigments, chlorophyll stability index, leaf RWC (relative water content), leaf soluble sugars, and many other growth-related parameters [23,24,25,26]. Furthermore, exogenous cytokinin spray ameliorates oxidative stresses by increasing the activities of antioxidant enzymes [24,26] and suppresses ABA-induced stomatal closure [27,28] during drought stress. Foliar spray of CPPU/Forchlorfenuron, a phenyl urea-based synthetic cytokinin, has been widely used in recent times for exogenous cytokinin treatment. CPPU acts as a competitive inhibitor of cytokinin oxidase/dehydrogenase (CKX), which allows plants to retain higher concentrations of cytokinin [29,30]. CPPU has also been implicated in increasing the weight and size of fruit and vegetable crops [31,32,33,34,35]. CPPU treatment in papaya promoted drought resistance by enhancing chlorophyll contents and antioxidant activities under drought stress conditions [36]. Our previous studies showed that the application of 5 mg/L CPPU promoted lateral branching, enhanced sugar contents, and the production of andrographolide compounds in a medicinal plant, *Andrographis paniculata* [37]. Another study proved that foliar spray of CPPU enhanced the salt tolerance in rice by maintaining the rate of photosynthesis, soluble sugars, and free proline concentration under salinity stress [16]. In the context of enhanced photosynthetic ability displayed by different crop plants in response to synthetic cytokinins, we have unraveled the cytokinin mediated proteomic changes corresponding to the process of photosynthesis. The rate of photosynthesis is directly influenced by stomatal conductance that ensures the availability of CO_2_ and the contents of chlorophyll pigments to harvest light energy. Henceforth, the present research focuses on investigating the effect of external cytokinin spray on stomatal conductance, photosynthetic pigments, and finally the rate of photosynthesis at two different growth stages of rice plants under drought stress. To investigate the corresponding alterations at a molecular level, we also examined the abundance intensities of proteins associated with the above-mentioned determinants of photosynthesis.

## 2. Materials and Methods

### 2.1. Plant Material, CPPU and Drought Stress Treatments

*Oryza sativa* ssp. *Indica* cv. PTT1 seeds were procured from the Laboratory of Plant Physiology and Agri-biotechnology, Faculty of Science, Mahidol University, Bangkok. PTT1 is a drought sensitive cultivar of rice [38]. The seeds were disinfected by sodium hypochlorite (Chlorox^®^ 10% *v*/*v*) for 30 min in a 100 mL flask, and were subsequently germinated in a container on moistened filter paper (Whatman^®^ no. 1) for 14 days in light at room temperature. Thirty seedlings (5 cm long, with true leaves) were transplanted with the spacing of 5 × 5 inch into the experimental blocks (length x width x height = 3 m × 2 m × 30 cm) filled with sand and soil (2:1) at the greenhouse, Salaya Campus, Mahidol University. Experimental blocks were supplemented with working Yoshida solution regularly to balance essential nutrient contents. Tensiometers (Soil Moisture, USA) were installed in the experimental blocks to keep a check on the soil moisture tension. Drought treatment was executed by withholding water for 14 days, whereas sufficient soil moisture (soil moisture tension was −15 kPa) was maintained for control well-watered plants. Synthetic cytokinin treatment was given exogenously by foliar spraying the plants with a 5 mg L^−1^ solution of CPPU (Kyowa Hakko Kogyo Co., Ltd.) at the rate of 25 mL/plant [37]. CPPU solution was added with 0.1% Tween 20^®^ that was used as a leaf surfactant. Control plants were sprayed with sterile water added with 0.1% Tween 20^®^ solution at the same time. Plants were sprayed with the CPPU solution only once on day 6 of drought stress (soil moisture tension −55 kPa). Samples were collected in three biological replicates to examine morphological, physiological, and biochemical changes on day 7 (soil moisture tension −60 kPa) and day 14 (soil moisture tension −72 kPa) of the drought stress treatment. A total of three treatments were used in the experiment: 1. Well-watered (WW) plants, 2. Drought stressed (DR) plants, and 3. Drought stressed plants with 5 mg/L CPPU (DR-CPPU). The experiments were conducted separately during two different growth stages of rice viz. tillering stage (60 days after germination) and grain-filling stage (75 days after germination). All the physiological, biochemical, and proteomic investigations were conducted on flag leaves with three biological replications.

### 2.2. Analysis of Stomatal Conductance and Net Photosynthetic Rate

The net photosynthetic rate (µmol m^−2^ s^−1^) and stomatal conductance (mmol m^−2^ s^−1^) were measured in flag leaves using a LI−6400XT Portable Photosynthesis System (LI-COR Biosciences). Measurements were taken in biological triplicates during late morning hours (10:30–11:30) using the following reference IRGA (infrared gas analyzer) chamber settings: CO_2_ Mixer: CO_2_R = 400 µmL, Coolers: Tblock = 28.0 °C, Flow: Fixed = 500 µmol s^−1^, Lamp: ParIn = 1000 µmL.

### 2.3. Spectrophotometric Analysis of Photosynthetic Pigments

Photosynthetic pigments (chlorophyll a and chlorophyll b and total carotenoids) were analyzed in biological triplicates according to the method of Wellburn [39]. About 0.1 g of fresh flag leaf sample was cut into pieces and homogenized with 5 mL of 80% acetone and kept at 4 °C for 48 h. The extract was filtered through a Whatman^®^ filter paper into a separate test tube. One mL of filtered extract was used for spectrophotometric determination (GENESYS™ 10S UV-Vis Spectrophotometer) of Chl a (chlorophyll a), Chl b (chlorophyll b), and carotenoids at 663, 645, and 470 nm absorbance, respectively. The quantification of Chl a, Chl b, and carotenoids was performed by the following standard equations: chlorophyll a = 12.7 A_663_–2.69A_645_, chlorophyll b = 22.9 A_645_–4.68A_663_, carotenoids = (1000 A_470_−2.270 Chl a–81.4 Chl b)/227. The results were expressed as micrograms of chlorophylls or carotenoids per gram of fresh leaf tissue (µg g^−1^).

### 2.4. Protein Extraction and Sample Preparation for Shotgun Proteomics

Fresh flag leaf samples were collected in triplicates (biological) from all of the treatment combinations of drought and CPPU for differential proteomic analysis. Proteins were extracted using a modified version of the protocol described by Shen [40]. Briefly, 100 mg of tissue was ground to a fine powder in liquid nitrogen and homogenized in pre-cooled 1 mL TCA extraction buffer (10% TCA in 100% acetone added with 0.07% fresh 2-mercaptoethanol). Samples were vortexed, incubated at −20 °C for 1 h, and centrifuged at 12,000 rpm for 5 min at 4 °C. Supernatants were discarded and precipitates were washed three times with ice-cold acetone solution (acetone containing 0.07% 2-mercaptoethanol). The precipitates were dried in the oven at 55 °C for 30 min, dissolved with a lysis buffer (30 mM Tris-base (Tris hydroxymethyl aminomethane), 7 M urea, 2 M thiourea, 4% CHAPS, pH 8.5), and vortexed and centrifuged at 12,000 rpm for 15 min. Supernatants containing crude protein mixtures were collected after centrifugation and stored at −20 °C. The concentration of proteins was measured using BSA (bovine serum albumin) as a standard protein [41] and absorbance was taken by Microplate Reader-TECAN (Spark 10M) at 595 nm. 10 µg protein samples from each biological triplicate were mixed for further LC-MS analysis. To reduce the disulfide bond, 10 mM dithiothreitol in 10 mM ammonium bicarbonate was added to the protein solution and reformation of disulfide bonds in the proteins was blocked by alkylation with 30 mM iodoacetamide in 10 mM ammonium bicarbonate. The protein samples were digested with sequencing grade porcine trypsin (1:20 ratio) for 16 h at 37 °C. The tryptic peptides were dried using a speed vacuum concentrator and re-suspended in 0.1% formic acid for nano-liquid chromatography-tandem mass spectrometry (nanoLC-MSMS) analysis.

### 2.5. Liquid Chromatography-Tandem Mass Spectrometry (LC/MS) and Data Analysis

Tryptic peptide samples were injected in triplicate into a HCTUltra LC-MS system (Bruker Daltonics Ltd.; Hamburg, Germany), coupled with a nanoLC system: UltiMate 3000 LC System (Thermo Fisher Scientific; Madison, WI, USA) as well as an electrospray at the flow rate of 300 nL-min^−1^ to a nanocolumn (PepSwift monolithic column 100 mm internal diameter 50 mm). Mobile phases consisting of solvent A (0.1% formic acid) and solvent B (80% acetonitrile and 0.1% formic acid) were used to elute peptides using a linear gradient of 10–70% of solvent B at 0–13 min (the time-point of retention), followed by 90% B at 13–15 min to transfer all peptides in the column. The final elution of 10% B at 15–20 min was carried out at the end to remove any remaining salt. The quantitation of LC-MSMS data was performed by Differential Analysis software (DeCyderMS, GE Healthcare) [42,43], and the identification of proteins was performed by searching against the Oryza sativa non-redundant subset database of the National Center for Biotechnology Information (NCBI). The mass spectrometry proteomics data have been deposited to the ProteomeXchange Consortium via the PRIDE [1] partner repository with the dataset identifier PXD021005. Searches were performed with a maximum of three missed cleavages, carbamidomethylation of Cys as a fixed modification, and oxidation of Met as variable modifications. Protein scores were derived from ion scores as a nonprobabilistic ranking protein hits and obtained as the sum of peptide scores. Data normalization and the quantification of the changes in protein abundance were performed among different treatments by MultiExperiment Viewer (MeV) in the TM4 suite software [44]. The relative abundance of peptides was presented as log2 abundance intensities. The highest log2 abundance intensity value among the three technical replicates was used as the representative value of that treatment.

## 3. Results and Discussion

### 3.1. Synthetic Cytokinins Improve Stomatal Conductance during Drought Stress

Stomata are the environmentally controlled gateways in the plants for CO_2_ uptake and transpiration, and therefore play a vital role in determining the rate of photosynthesis [45]. In response to water deficit stress, plants need to meticulously balance the CO_2_ uptake and water transpiration through the stomatal aperture. Like various other stress induced adaptations, plants under drought stress conventionally follow the strategy to reduce the water loss through transpiration by closing stomatal apertures [46]. Cytokinins at high concentrations have been reported to revert the ABA-induced stomatal closure [27,28] during abiotic stresses. In our study, we evaluated the effect of externally applied synthetic cytokinin on the stomatal conductance under drought stress (Figure 1).

There was an obvious effect of drought stress on the stomatal closure, leading to reduced conductance at both tillering and grain-filling stages. Conversely, the well-watered plants maintained a healthy stomatal conductance. However, CPPU had a noticeable impact of on stomatal conductance under drought stress, whereby the stomatal conductance of CPPU treated plants was significantly higher than that of the untreated plants, with an exception on day 14 at tillering stage.

In order to authenticate the effect of drought and CPPU on stomatal conductance, we investigated the abundance intensities of the proteins (immediately extracted from the same tissues) that were either directly or indirectly related to the stomatal conductance. During drought stress, ABA concentration and signaling plays a vital role in controlling the stomatal conductance. ABA is synthesized in the roots and leaves, and transported to the guard cells via ATP-binding cassette (ABC) transporters that are located in the plasma membrane [46]. After reaching the guard cells, ABA and its signaling components modulate the ion channel activities including the efflux of anions and potassium ions and the inhibition of K^+^ import that leads to the closure of stomata [47]. To examine the involvement of ABA and its signaling components in controlling the stomatal conductance under drought stress, we evaluated the abundance intensities of some important proteins that regulate the biosynthesis and degradation of ABA. Zeaxanthin epoxidase (ZEP) and 9-cis-epoxycarotenoid dioxygenase (NCED) proteins are implicated in the biosynthesis of ABA [48], whereas Abscisic acid 8′-hydroxylase is responsible for the oxidative degradation of ABA [49]. In our study, both the proteins involved in ABA biosynthesis, ZEP and NCED, were relatively more abundant in the drought stressed plants at both day 7 and 14 of drought stress (Figure 2). In contrast, the CPPU treated plants under drought stress retained the normal levels of NCED proteins, similar to well-watered plants. However, the abundance of ZEP in synthetic cytokinin treated plants looked similar to the levels in drought stressed plants at the tillering stage. Abscisic acid 8′-hydroxylase protein, on the other hand, had equal abundance in all the treatments during both the growth stages of the rice. The results suggest that the treatment of synthetic cytokinin helped the plants to confine the drought induced biosynthesis of ABA and thereby facilitated the improved stomatal conductance. Further, the degradation of ABA was not swayed by either the drought or the synthetic cytokinin treatment.

ABA signaling is umpired by its receptors, namely PYR/PYL/RCARs (Pyrabactin resistance/Pyrabactin-like/Regulatory components of the ABA receptor) and SnRK (Sucrose non-fermenting−1-related protein kinase) protein kinases, which phosphorylate the downstream targets and trigger the ABA-induced responses in plants. Under lesser availability of ABA, the function of SnRK2 is subdued by PP2C (Protein phosphatase type−2C) phosphatases, which act as negative regulators of ABA signaling. When the concentration is higher, ABA binds to its receptors, which in turn bind to PP2Cs and inactivate them. Consequently, PP2Cs are dissociated from SnRK2s, resulting in the activation of SnRK2s to initiate ABA-induced responses [3,50,51]. The ABA receptors, PYL9 and PYL10, have been independently reported for their roles in leaf senescence, lateral root elongation [52], and drought and cold tolerance [53,54]. In our study, the abundance of both the receptor proteins, PYL9 and PYL10, was significantly induced under drought stress (Figure 3), and there was no effect of CPPU treatment on the abundance intensity of PYL10 under drought stress. Nevertheless, abundance of PYL9 under drought stress was intimidated by CPPU treatment at both the tillering and grain-filling stages of the rice. The results revealed a robust drought inducible character of both PYL9 and PYL10 receptors. Among other downstream proteins of ABA signaling, SnRK4 and ABA insensitive (ABI) 5 showed differential abundance in response to synthetic cytokinin application under drought stress. An abundance of SnRK4 and ABI5 proteins was induced by drought stress in our study (Figure 4). However, CPPU treated plants under drought stress maintained normal abundance levels of these proteins, which was comparable with well-watered plants. ABI4 and ABI5 proteins act as positive regulators of ABA signaling ABA arbitrated responses in plants [55,56]. Some other downstream proteins involved in ABA signaling like PP2C04, BIPP2C1, ABA insensitive (ABI) 4, and SnRK1 showed equal abundance in all the treatments without being influenced by drought or the CPPU treatment (Table 1).

The opening and closing of stomata is tightly regulated through various ion channels located in the guard cell membranes. During the opening of the stomata, the H^+^-ATPase pumps facilitate the efflux of H^+^ from the guard cells. Extrusion of H^+^ ions from the guard cells results in the acidification of the apoplast that leads to K^+^ uptake via activation of inward potassium channels [57,58]. The potassium (K^+^) channels were first discovered in *Arabidopsis thaliana* and have been named as ATKs [59,60]. Phototropins (phot1, phot2) have been widely reported to control the stomatal opening through the activation of plasma membrane bound H^+^-ATPase [61]. Localized on the outer membrane of chloroplast, phototropins are the plant-specific protein kinases that act as blue light photoreceptors [62]. They regulate a wide range of physiological processes such as stomatal opening, chloroplast relocations, and phototropism (bending towards light) in order to maximize the photosynthetic efficiency [63,64,65]. Genetic analysis has revealed two different phototropins (Phot 1 and 2) with partially overlapping functions in plants [66]. BLUS1 (BLUe light Signaling 1), a Ser/Thr protein kinase, mediates the primary step for phototropin signaling in guard cells. Phototropins phosphorylate the Ser-348 residue within C-terminus of BLUS1 and trigger its kinase activity, which subsequently phosphorylates and activates the plasma membrane H^+^-ATPase, causing stomata to open [61,67]. We investigated the effect of CPPU on the abundance intensities of both the phototropins (Phot1 and Phot2) under drought stress (Figure 5), whereby both blue light receptor kinases were substantially intimidated by drought stress at tillering and grain-filling stages. However, CPPU treatment helped the plants retain the normal levels of these blue light receptor kinases under drought stress like the well-watered plants. Potassium channels (KAT1, 2, 5, and 6), located on the plasma membrane of guard cells, also have key roles in stomatal opening as they facilitate the inward uptake of K^+^ ions into the guard cells [59,60]. In our study, we also analyzed the abundance of these potassium channel proteins (KAT1, 2, 5, and 6) under different treatments in rice. However, they remained unaffected across all of the treatments in rice (Table 1).

In our study, the abundance intensities of proteins were not substantially influenced by the growth stages (tillering and grain-filing) of rice. Effects of drought and CPPU were predominantly noticeable on the proteins involved in ABA biosynthesis and signaling and phototropins. ABA biosynthesis and signaling proteins, indorsing stomatal closure, were induced under drought stress that might have caused stomatal closure and reduced the stomatal conductance in drought stressed plants. In contrast, CPPU treatment curtailed the effect of drought on these proteins, resulting in improved stomatal conductance under drought stress. Furthermore, phototropins (Phot1 and Phot2), implicated in stomatal opening, were suppressed by drought stress, which might have resulted in the partial closure of stomata and poor stomatal conductance in drought stressed plants. Interestingly, CPPU treated plants under drought stress retained the higher levels of phototropins, which was comparable with the well-watered plants, which might have resulted in improved stomatal conductance.

### 3.2. Synthetic Cytokinins Augment Chl b and Confine Carotenoid Contents under Drought Stress

Photosynthetic pigments are essential for plants to harvest light energy and produce reducing powers. Pigments are prone to environmental stresses, particularly drought (Farooq et al., 2009). Osmotic stresses reduce chlorophyll b contents without substantially affecting the contents of Chl a in drought stressed leaves [68,69]. To apprehend the influence of CPPU on photosynthetic pigments under drought stress, we quantified Chl a and Chl b spectrophotometrically in drought stressed leaves of rice (Figure 6). Chl a contents were largely unaffected by either CPPU treatment or drought stress and the concentration fluctuated in a very narrow range of 26–32 µg g^−1^. Drought stress perpetually reduced the Chl b contents in the leaves at both growth stages of the rice. Chl b contents increased significantly in response to CPPU treatment under drought stress conditions. Interestingly, the Chl b content of CPPU treated plants under drought stress was higher than that in well-watered plants at day 14 in the tillering stage and at day 7 and day 14 in the grain-filling stage.

To authenticate the effect of drought and synthetic cytokinin spray on photosynthetic pigments, we investigated the abundance intensities of two key enzymes, chlorophyll synthase and 7-hydroxymethyl chlorophyll a reductase (HCAR), involved in biosynthesis/degradation of chlorophyll pigments. Chlorophyll synthase, involved in the final step of the biosynthetic pathway of chlorophylls, catalyzes the esterification of chlorophillide a or b with phytyl or geranyl-geranyl pyrophosphate into chlorophyll a or b as final product in the chloroplast [70,71]. Reduced expression of chlorophyll synthase also instigates a feedback-controlled inactivation of the initial and rate-limiting step of the chlorophyll synthetic pathway in plants [72]. In our study, the abundance of chlorophyll synthase protein was severely affected by the drought exposure, whereas CPPU treated plants retained sufficient levels of this enzyme under drought stress (Figure 7A). Another enzyme involved in chlorophyll metabolism, HCAR, revealed contrasting abundance patterns under drought stress (Figure 7B). It showed enhanced abundance in drought stressed leaves and relatively low abundance in well-watered and CPPU treated plants under drought stress. HCAR belongs to the iron-sulfur flavoprotein group containing FAD and an iron-sulfur center [73]. It catalyzes the reduction of a hydroxymethyl group to a methyl group and thereby converts 7-hydroxymethyl chlorophyll to chlorophyll a in the chloroplast. Interestingly, 7-hydroxymethyl chlorophyll, which serves as substrate for HCAR, is derived from the NaBH_4_ mediated reduction of chlorophyll b in the chloroplast [74]. Henceforth, HCAR has been categorized as a catabolic enzyme of the chlorophyll cycle that carries out the second and the last step in the conversion of chlorophyll b to chlorophyll a [75]. Previous studies in Arabidopsis and rice indicated that HCAR-overexpressing plants exhibited accelerated leaf yellowing and senescence due to the degradation of chlorophyll b in the chloroplast, whereas HCAR knockout mutants exhibited persistent green leaves during both dark-induced and natural senescence [76,77]. The abundance intensities of chlorophyll synthase and HCAR proteins clearly corroborate with the results of spectrophotometrically quantified Chl a and Chl b pigments. Consequent to the reduced abundance of chlorophyll synthase, it can be hypothesized that drought stressed plants maintain the concentration of Chl a at the cost of Chl b through enhanced levels of HCAR, which actively converts Chl b into Chl a. On the other hand, synthetic cytokinin treatment helps sustain the expression of chlorophyll synthase under drought stress, which in turn helps retain the contents of chlorophyll pigments (both Chl a and Chl b) at satisfactory levels, which is comparable with well-watered plants.

Carotenoid contents tend to increase under osmotic stresses as they have additional roles during stress conditions and partially help the plants to withstand drought [78,79]. Total carotenoid contents were also measured spectrophotometrically in all of the treatments (Figure 8A,B). Drought stressed plants, in most cases, retained higher contents of carotenoids compared with well-watered plants. However, with the progression of drought stress, from day 7 to day 14, carotenoid contents invariably demonstrated a severe reduction in all of the treatments. However, CPPU treatment helped in moderating carotenoids contents under drought stress. To verify the influence of drought and synthetic cytokinin on the contents of total carotenoids, we investigated the abundance intensities of Carotenoid cleavage dioxygenases 8 (CCD8) in all of the treatments (Figure 8C). Carotenoid cleavage dioxygenases are responsible for the oxidative cleavage of carotenoids [80]. CCD8 in particular is involved in strigolactones biosynthesis by cleaving the C(27) 9-cis-10′-apo-beta-carotenal produced by CCD7 [81]. Strigolactones are hormones that inhibit tillering and shoot branching through the MAX-dependent pathway [82] and have been widely implicated in acclimation to environmental stresses [83,84]. Recent studies suggest that CRISPR/Cas9-mediated mutagenesis of CCD8 alters the root and shoot architecture and provides resistance against some parasitic weeds [85,86,87]. In our study, CCD8B protein was induced by CPPU during severe drought stress (day 14) at both the tillering and grain-filling stages. The results largely corroborated with the concentration of carotenoids in the synthetic cytokinin treated plant, particularly at the grain-filling stage under drought stress. Evidently, the concentration of total carotenoids was considerably less in CPPU treated plants on day 14 of drought stress at both tillering and grain-filling stages of the rice. The inclusive results of photosynthetic pigments implied the positive effect of synthetic cytokinin treatment on the total chlorophyll contents (Chl a + Chl b) during drought stress, which may be corroborated with the proteomic patterns where the abundance intensities of chlorophyll synthase and HCAR enzymes were meticulously controlled by synthetic cytokinin under drought stress. Furthermore, low contents of carotenoids in CPPU treated plants might be comprehended by the abundance of CCD8B enzyme in their leaves.

### 3.3. Synthetic Cytokinins Stimulate Rubisco Activity and Uphold the Rate of Photosynthesis during Drought Stress

The rate of photosynthesis is invariably reduced under osmotic stress conditions primarily due to the closure of stomata, which hampers CO_2_ intake [88]. Furthermore, drought induced decline in chlorophyll pigments also accounts for the reduced rate of photosynthesis [79]. The role of externally applied synthetic cytokinins in improving the photosynthetic rate under environmental stress conditions has been reported in various crop species [14,15,16,17]. After investigating the effect of synthetic cytokinin treatment on the stomatal conductance and photosynthetic pigments during drought stress, it was pertinent to examine the cumulative effect of these factors on the rate of photosynthesis. Henceforth, we investigated the effect of synthetic cytokinin treatment on the net photosynthetic rate in drought stressed plants (Figure 9). The results indicated that the net photosynthetic rate was perpetually higher in well-watered plants irrespective of CPPU treatment. On the other hand, the net photosynthetic rate was drastically reduced by drought stress at both the tillering and the grain-filling stages. Nevertheless, CPPU treated plants maintained a significantly higher rate of photosynthesis compared with untreated plants under drought stress. At the tillering stage, the adverse effect of drought on photosynthesis was virtually identical on day 7 and day 14 of drought stress. On the other hand, there was a steep decrease in the rate of photosynthesis on day 14 of drought stress in the plants at the grain-filling stage.

To verify the positive impact of synthetic cytokinin treatment on the process of photosynthesis under drought stress, we investigated the abundance intensities of two regulatory proteins, Rubisco accumulation factor 1 (Raf1) and Ribulose bisphos carbo/oxygenase (Rubisco) activase, required for the proper functioning of Rubisco. Rubisco, a complex of eight large (RbcL) and eight small (RbcS) subunits, is the rate-limiting carbon-fixing enzyme of photosynthesis. Rubisco activase and Raf1 are the key enzymes that regulate the activity of Rubisco and consequently the process of photosynthesis. Rubisco activase (Rca) acts as a catalytic chaperone in regulating the activity of Rubisco by facilitating the dissociation of inhibitory sugar phosphates from the active site of Rubisco in an ATP-dependent manner [89,90]. On the other hand, Raf1 is a key molecular chaperone, which assists the assembly of the Rubisco subunits [91,92]. Raf1 dimers facilitate the stabilization and assembly of the post-chaperonin-folded Rubisco L-subunits [93]. In our study, abundance intensities of both Rubisco activase and Raf1 proteins was intimidated by drought stress at the tillering and grain-filling stages of the rice (Figure 10). However, the synthetic cytokinin treatment helped the plants sustain the normal levels of these proteins, thereby assisting plants to uphold the rate of photosynthesis during drought stress. The synthetic cytokinin treatment also induced phototropins (Phot1 and Phot2), which are implicated in blue light reception and chloroplast translocation to assist the process of photosynthesis.

## 4. Conclusions

The three major external variables that precisely control the process of photosynthesis are light, CO_2_, and water. Water deficit stress severely affects the rate of photosynthesis in plants, which leads to leaf wilting, reduced fresh biomass, and reduced yield. During the process of photosynthesis, the availability of CO_2_ is ensured by optimal stomatal conductance, whereas the photosynthetic pigments assist in harvesting the light energy. The ubiquitous enzyme, Rubisco, fixes atmospheric CO_2_ into the energy-rich carbon molecules. Cytokinins help plants sustain normal growth and development by positively modulating various drought induced morphological, physiological, and biochemical processes. Enhanced cytokinins in plants induce the expression of phototropins implicated in stomatal opening and curtail ABA biosynthesis and signaling, thus improving the stomatal conductance in plants during drought stress. Cytokinins control the biosynthesis of chlorophyll pigments meticulously by increasing the abundance of chlorophyll synthase and confining the abundance of HCAR under drought stress. Further, cytokinins modulate the expression of key proteins involved in the assembly and activation of the Rubisco enzyme, thus helping maintain the photosynthesis at a decent rate during drought stress. Conclusively, synthetic cytokinins reverse the drought induced alterations in the plants and allow normal growth and developmental activities (Figure 11). Plenty of research findings suggest that the cytokinins support normal growth and development under osmotic stresses and improve the drought tolerance ability of plants. However, the precise molecular mechanism of cytokinin mediated drought tolerance is yet to be discovered.

## Figures and Tables

**Figure 1 plants-09-01106-f001:**
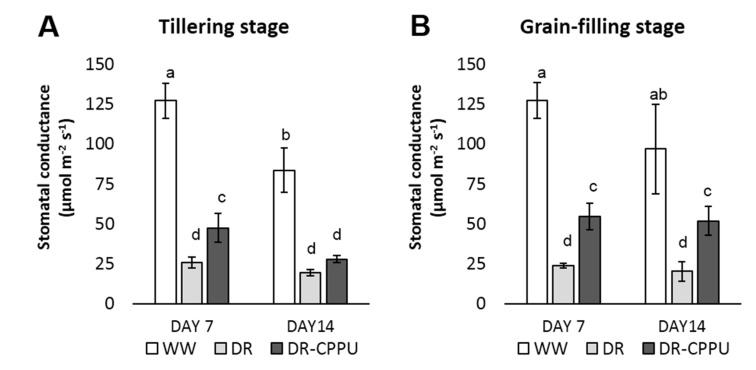
Stomatal conductance (mmol m^−2^ s^−1^) of rice leaves under different treatment conditions at tillering (**A**) and grain-filling (**B**) stages. WW = Well-watered plants (soil moisture tension of −15 kPa), DR = Drought stressed plants (Soil moisture tension of −55 kPa and −72 kPa at day 7 and day 14, respectively), and DR-CPPU = Drought stressed plants, sprayed with 5 mg/L CPPU on day 6 of drought treatment. Error bars represent SD (standard deviation). Letters viz. a, b, c, d, e, over SD bars indicate significant differences of mean at *p* < 0.05 (*), as analyzed by Duncan’s Multiple Range Test (DMRT).

**Figure 2 plants-09-01106-f002:**
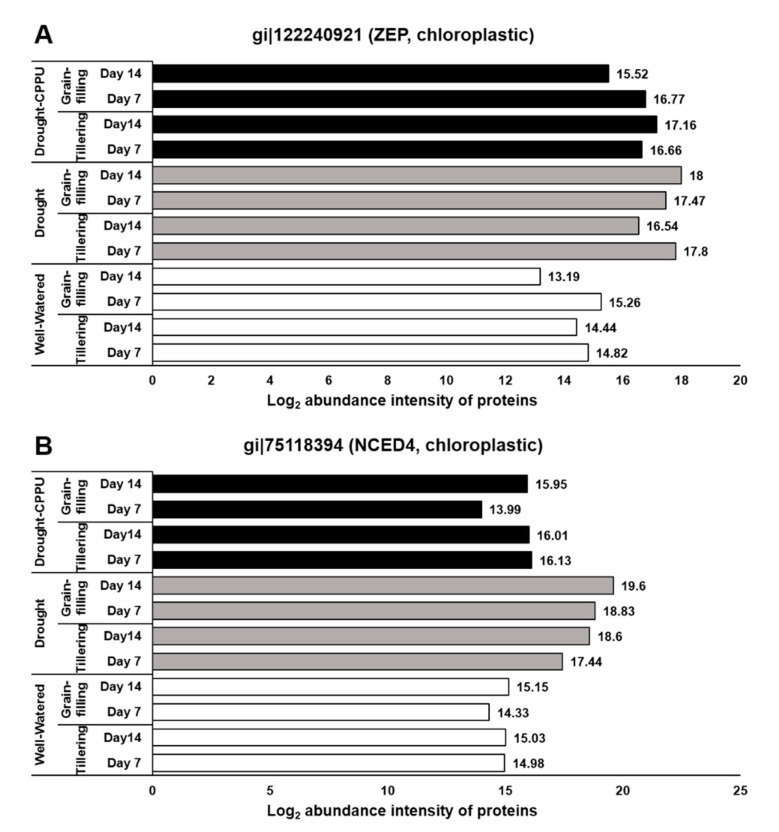
The influence of synthetic cytokinin and drought on the abundance intensities of ZEP (**A**) and NCED4 (**B**) proteins, implicated in ABA biosynthesis and stomatal conductance, at the tillering and grain-filling stages of the rice. Protein abundance intensities, represented on the X-axis, are the highest log2 fold change values of technical replicates. [Well-Watered plants were maintained at a soil moisture tension of −15 kPa during the treatment period. Drought stress was imposed by withholding water for up to 14 days. Soil moisture tensions of −55 kPa and −72 kPa were recorded at day 7 and day 14 during the treatment, respectively. CPPU treatment was given by foliar spraying plants with a 5 mg/L solution of CPPU at the rate of 25 mL/plant on day 6 of drought treatment.

**Figure 3 plants-09-01106-f003:**
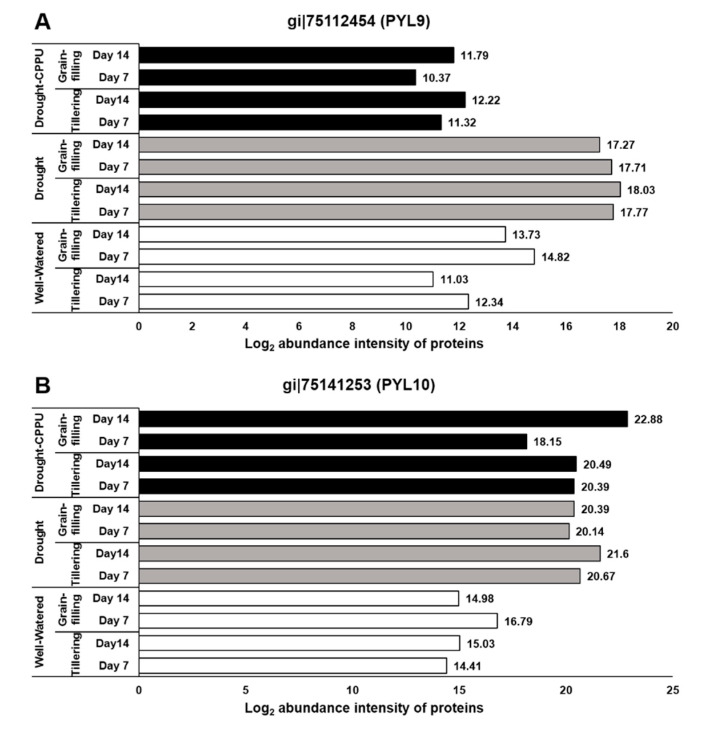
The influence of synthetic cytokinin and drought on the abundance intensities of PYL9 (**A**) and PYL10 (**B**) proteins, implicated in ABA reception and stomatal conductance, at the tillering and grain-filling stages of the rice. Protein abundance intensities, represented on the X- axis, are the highest log2 fold change values of technical replicates. [Well-Watered plants were maintained at a soil moisture tension of −15 kPa during the treatment period. Drought stress was imposed by withholding water for up to 14 days. Soil moisture tensions of −55 kPa and −72 kPa were recorded at day 7 and day 14 during the treatment, respectively. CPPU treatment was given by foliar spraying plants with a 5 mg/L solution of CPPU at the rate of 25 mL/plant on day 6 of drought treatment.

**Figure 4 plants-09-01106-f004:**
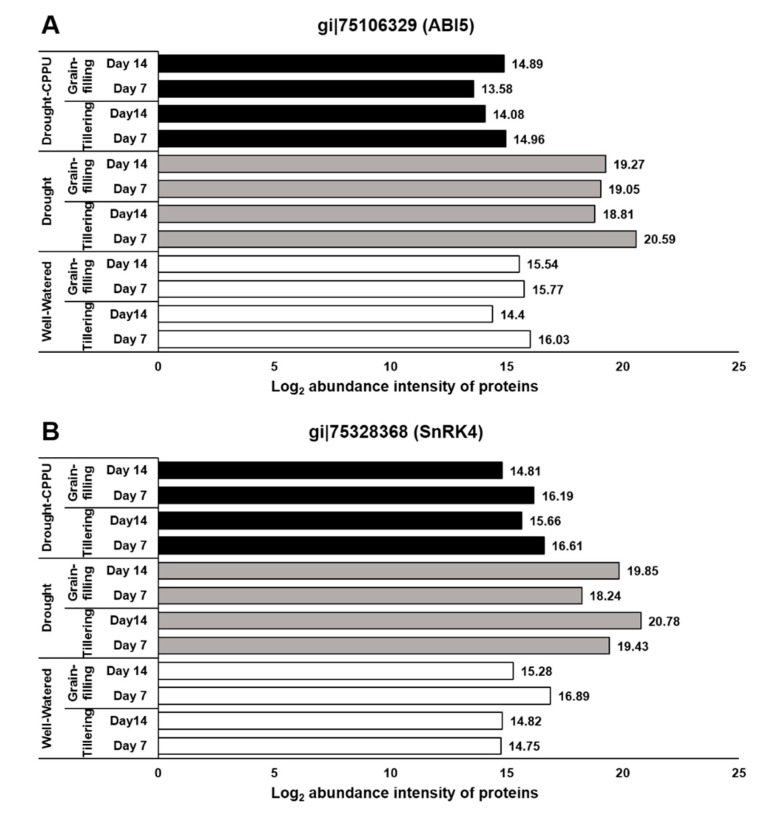
The influence of synthetic cytokinin and drought on the abundance intensities of ABI5 (**A**) and SnRK4 (**B**) proteins, implicated in ABA signaling and stomatal conductance, at the tillering and grain-filling stages of the rice. Protein abundance intensities, represented on the X- axis, are the highest log2 fold change values of technical replicates. [Well-Watered plants were maintained at a soil moisture tension of −15 kPa during the treatment period. Drought stress was imposed by withholding water for up to 14 days. Soil moisture tensions of −55 kPa and −72 kPa were recorded at day 7 and day 14 during the treatment, respectively. CPPU treatment was given by foliar spraying plants with a 5 mg/L solution of CPPU at the rate of 25 mL/plant on day 6 of drought treatment.

**Figure 5 plants-09-01106-f005:**
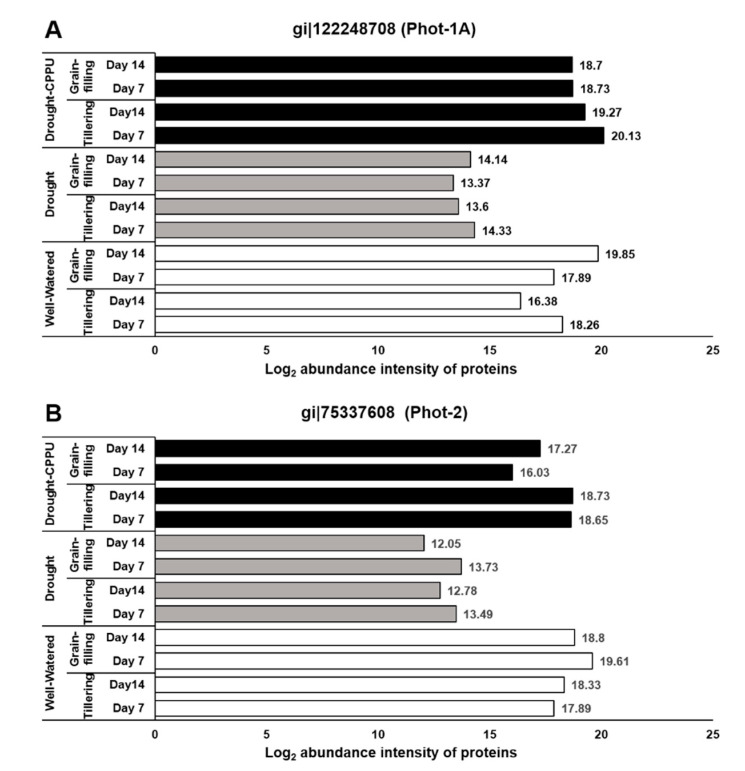
The influence of synthetic cytokinin and drought on the abundance intensities of phototropins, Phot-1 (**A**), and Phot-2 (**B**) proteins, implicated in photosynthesis and stomatal conductance at tillering and grain-filling stages of the rice. Protein abundance intensities, represented on the X- axis, are the highest log2 fold change values of technical replicates. [Well-Watered plants were maintained at a soil moisture tension of −15 kPa during the treatment period. Drought stress was imposed by withholding water for up to 14 days. Soil moisture tensions of −55 kPa and −72 kPa were recorded at day 7 and day 14 during the treatment, respectively. CPPU treatment was given by foliar spraying plants with a 5 mg/L solution of CPPU at the rate of 25 mL/plant on day 6 of drought treatment.

**Figure 6 plants-09-01106-f006:**
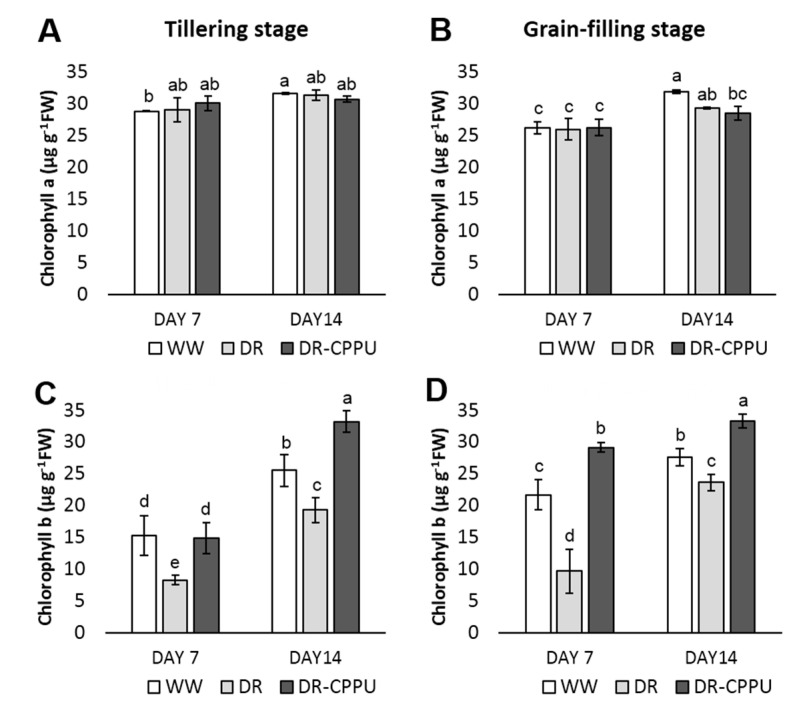
Contents of chlorophyll pigments (µg g^−1^ FW), Chl a (**A**,**B**) and Chl b (**C**,**D**), quantified from rice leaves under different treatment conditions at tillering and grain-filling stages. WW = Well-watered plants (soil moisture tension of −15 kPa), DR = Drought stressed plants (Soil moisture tension of −55 kPa and −72 kPa at day 7 and day 14, respectively), and DR-CPPU = Drought stressed plants, sprayed with 5 mg/L CPPU on day 6 of drought treatment. Error bars represent SD (standard deviation). Letters viz. a, b, c, d, e, over SD bars indicate the highly significant differences of mean at *p* < 0.01 (**) as analyzed by Duncan’s Multiple Range Test (DMRT).

**Figure 7 plants-09-01106-f007:**
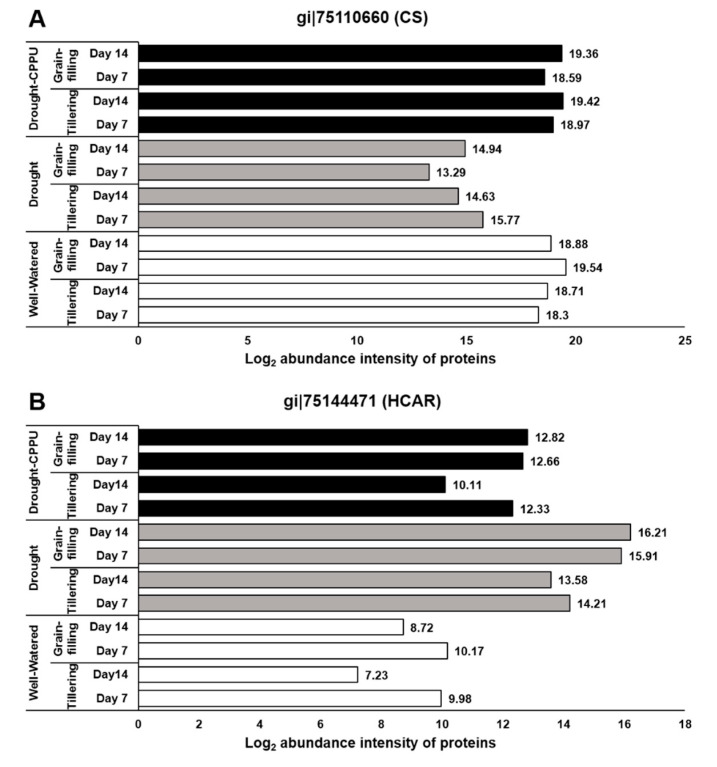
The influence of synthetic cytokinin and drought on the abundance intensities of (**A**) chlorophyll synthase (CS) and (**B**) 7-hydroxymethyl chlorophyll a reductase (HCAR) proteins involved in biosynthesis/degradation of chlorophyll pigments at tillering and grain-filling stages of the rice. Protein abundance intensities, represented on the X- axis, are the highest log2 fold change values of technical replicates. [Well-Watered plants were maintained at a soil moisture tension of −15 kPa during the treatment period. Drought stress was imposed by withholding water for up to 14 days. Soil moisture tensions of −55 kPa and −72 kPa were recorded at day 7 and day 14 during the treatment, respectively. CPPU treatment was given by foliar spraying plants with a 5 mg/L solution of CPPU at the rate of 25 mL/plant on day 6 of drought treatment.

**Figure 8 plants-09-01106-f008:**
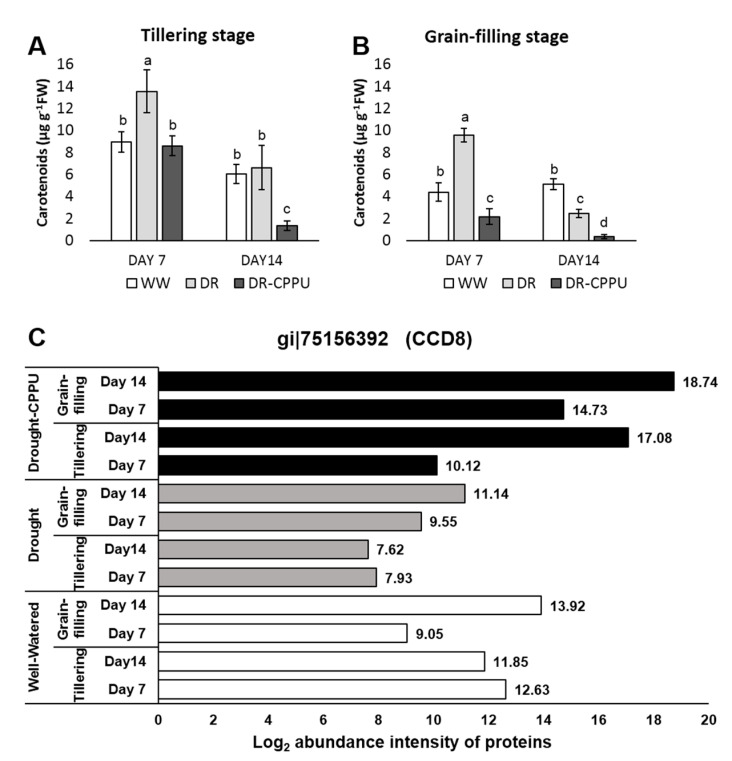
Total carotenoid contents (µg g^−1^ FW) quantified from rice leaves under different treatment conditions at tillering (**A**) and grain-filling (**B**) stages. WW = Well-watered plants (soil moisture tension of −15 kPa), DR = Drought stressed plants (Soil moisture tension of −55 kPa and −72 kPa at day 7 and day 14, respectively), and DR-CPPU = Drought stressed plants, sprayed with 5 mg/L CPPU on day 6 of drought treatment. Error bars represent SD (standard deviation). Letters viz. a, b, c, d, e, over SD bars indicate the highly significant differences of mean at *p* < 0.01 (**), as analyzed by Duncan’s Multiple Range Test (DMRT). (**C**) Influence of synthetic cytokinin and drought on the abundance intensities of the Carotenoid cleavage dioxygenases 8 (CCD8) protein involved in oxidative degradation of carotenoids at the tillering and grain-filling stages of the rice. Protein abundance intensities, represented on the X- axis, are the highest log2 fold change values of technical replicates. [Well-Watered plants were maintained at a soil moisture tension of −15 kPa during the treatment period. Drought stress was imposed by withholding water for up to 14 days. Soil moisture tensions of −55 kPa and −72 kPa were recorded at day 7 and day 14 during the treatment, respectively. CPPU treatment was given by foliar spraying plants with a 5 mg/L solution of CPPU at the rate of 25 mL/plant on day 6 of drought treatment.

**Figure 9 plants-09-01106-f009:**
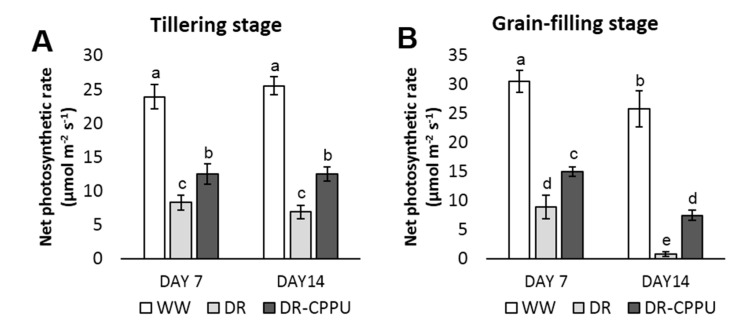
Net photosynthetic rate (µmol m^−2^ s^−1^), measured from rice leaves under different treatment conditions at the tillering (**A**) and grain-filling (**B**) stages. WW = Well-watered plants (soil moisture tension of −15 kPa), DR = Drought stressed plants (Soil moisture tension of −55 kPa and −72 kPa at day 7 and day 14, respectively), and DR-CPPU = Drought stressed plants, sprayed with 5 mg/L CPPU on day 6 of drought treatment. Error bars represent SD (standard deviation). Letters viz. a, b, c, d, e, over SD bars indicate the highly significant differences of mean at *p* < 0.01 (**), as analyzed by Duncan’s Multiple Range Test (DMRT).

**Figure 10 plants-09-01106-f010:**
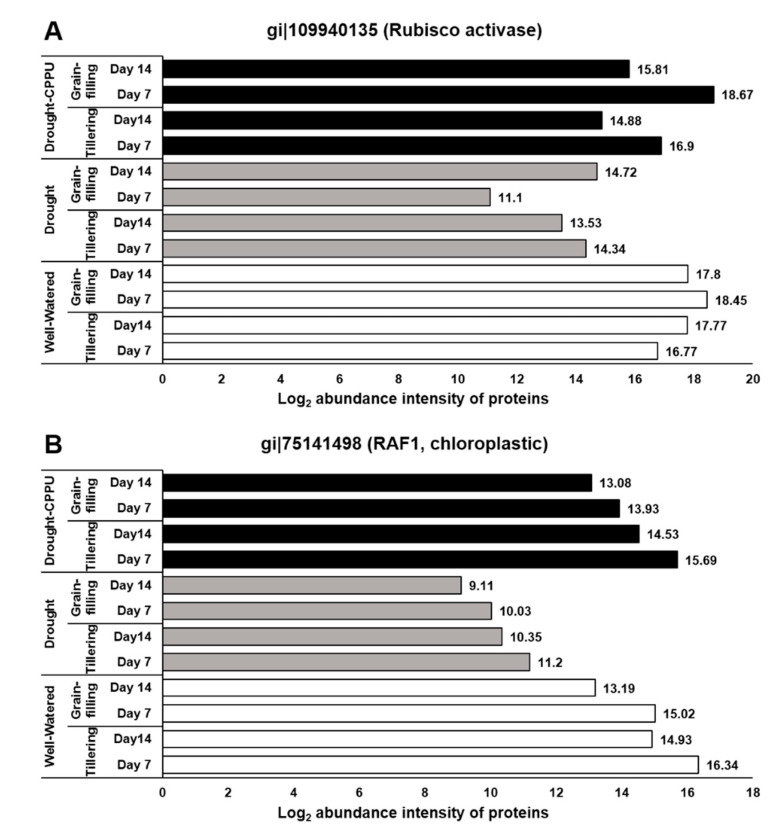
The influence of synthetic cytokinin and drought on the abundance intensities of (**A**) Ribulose bisphos carbo/oxygenase (Rubisco) activase and (**B**) Rubisco accumulation factor (Raf) 1 proteins involved in the regulation and functioning of Rubisco at the tillering and grain-filling stages of the rice. Protein abundance intensities, represented on the X- axis, are the highest log2 fold change values of technical replicates. [Well-Watered plants were maintained at a soil moisture tension of −15 kPa during the treatment period. Drought stress was imposed by withholding water for up to 14 days. Soil moisture tensions of −55 kPa and −72 kPa were recorded at day 7 and day 14 during the treatment, respectively. CPPU treatment was given by foliar spraying plants with a 5 mg/L solution of CPPU at the rate of 25 mL/plant on day 6 of drought treatment.

**Figure 11 plants-09-01106-f011:**
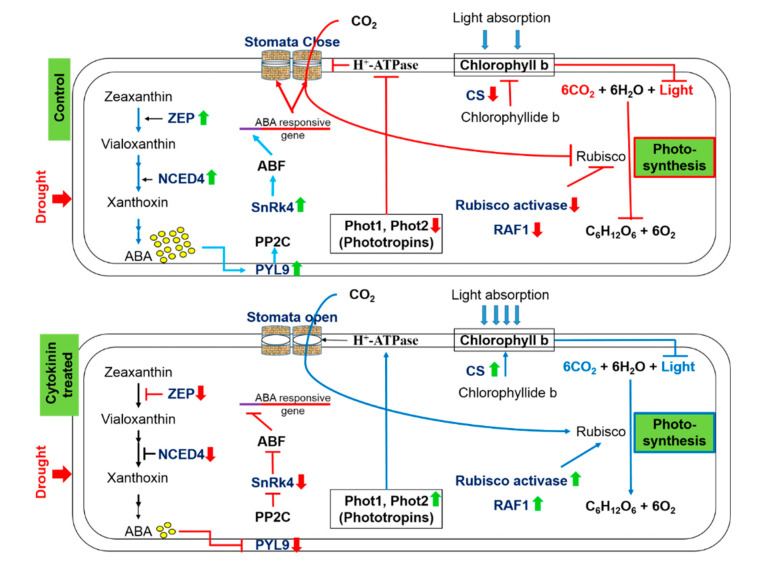
The positive impact of synthetic cytokinin on stomatal conductance, contents of chlorophyll b, and the process of photosynthesis under severe drought stress. Up (green) and down (red) arrows are the symbolic representations of increase and decrease, respectively. ZEP = Zeaxanthin epoxidase, NCED = 9-cis-epoxycarotenoid dioxygenase, ABA = Abscisic acid, PYL9 = Pyrabactin-like receptor 9, PP2C = Protein phosphatase type-2C, SnRK4 = Sucrose non-fermenting-1-related protein kinase 4, ABF = ABA responsive factor, Phot = Phototropin, CS = Chlorophyll synthase, RAF1 = Rubisco accumulation factor 1.

**Table 1 plants-09-01106-t001:** Rice flag leaf proteins (related to stomatal conductance) with unchanged abundance intensities in different treatments. The highest log2 intensity value among the three technical replicates was used as the representative value of that treatment.

GI No.	Name of Protein	Well-Watered ^1^				Drought ^2^				Drought-CPPU ^3^			
		Tillering		Grain-filling		Tillering		Grain-filling		Tillering		Grain-filling	
		Day 7	Day 14	Day 7	Day 14	Day 7	Day 14	Day 7	Day 14	Day 7	Day 14	Day 7	Day 14
gi|75124964	Potassium channel KAT1	16.43	14.47	14.42	15.89	15.94	16.13	15.62	14.27	14.64	16.22	16.12	15.45
gi|338810402	Potassium channel KAT2	17.77	18.55	17.13	18.80	19.33	18.59	17.68	18.49	18.80	18.33	20.57	18.47
gi|75144382	Potassium channel KAT5	16.90	18.63	17.25	17.08	18.90	17.51	17.83	17.36	18.25	17.09	17.11	16.33
gi|338810388	Potassium channel KAT6	15.79	17.97	16.92	14.16	14.91	15.32	16.37	16.72	16.65	15.82	16.23	15.93
gi|122163981	Abscisic acid 8′-hydroxylase 1	17.30	18.39	17.47	16.83	17.82	16.75	17.45	18.50	17.04	17.46	19.97	14.88
gi|75328369	Serine/threonine protein kinase OSK1 SnRK1	15.77	17.39	17.37	16.38	17.42	16.84	16.87	16.80	17.79	15.73	15.62	16.53
gi|75222723	Protein kinase and PP2C-like domain-containing protein PP2C04	14.60	16.37	15.36	16.13	17.05	17.19	16.43	16.52	16.71	17.15	15.32	14.86
gi|122247433	Protein phosphatase 2C BIPP2C1	16.70	19.26	16.88	16.48	17.46	17.21	17.78	16.50	18.91	17.80	18.52	18.68
gi|75123651	Protein ABIL4	20.64	19.89	20.49	18.95	21.65	19.52	20.20	20.21	17.72	18.42	20.44	19.66

^1^ Well-watered plants were maintained at a soil moisture tension of −15 kPa during the treatment period. ^2^ Drought stress was imposed by withholding water for up to 14 days. Soil moisture tensions of −55 kPa and −72 kPa were recorded at day 7 and day 14 during the treatment, respectively. ^3^ CPPU treatment was given by foliar spraying plants with a 5 mg/L solution of CPPU [N-2-(chloro-4-pyridyl)-N-phenyl urea] at the rate of 25 mL/plant on day 6 of drought treatment.

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
