# Peer review of "A Synthetic Cytokinin Improves Photosynthesis in Rice under Drought Stress by Modulating the Abundance of Proteins Related to Stomatal Conductance, Chlorophyll Contents, and Rubisco Activity"

_plants, 2020, doi:10.3390/plants9091106_

Round 1
Reviewer 1 Report
I have read the manuscript entitled: “A synthetic cytokinin improves photosynthesis in rice under drought stress by modulating the expression of proteins related to stomatal conductance, chlorophyll contents, and Rubisco activity” by Gujjar et al. In this research article the authors present the effect of cytokinin- like plant growth regulator on rice during water stress condition. Physiological parameters are analysed and protein abundances using MS approaches determined for some targeted proteins. Overall, the study is interesting, but the methods and presentation of the results should be improved for better clarity.
To allow for the clear understanding and reliability of the MS results the LCMS data should be presented as sup data and the full MS data fully accessible in MS repository databases.
There is a need for a figure that summarizes the results obtained, ie physiological and protein data.
The wording “expression” for proteins is not adequate, ‘abundance’ is a better term and should be corrected accordingly in the manuscript.
Methods should be more precise.
Line 92, Please add duration and if any washes done
Line 92. Was the paper left on the bench or in a container, with light or not, etc
Line 93. What is an “experimental block”?, ie length, depth, etc. what is the number of plants per block, ie density.
Line 102. Any tween is the control, if yes please add.
Line 121. What is small pieces?
Line 130. How many plants per replicate?
Line 135. Centrifugation steps at room temp?
Line 137. How long the drying?
Line 137. What was the volume/dried mass?
Figures with vertical bars have asterisks to indicate the significance level but already indicated in the legend and is a bit awkward in the figure itself, I would remove it.
Figures with protein abundance: Why there is no standard deviation presented while the methods indicate 3 replicates? The manuscript would benefit to have all those results presented in a single figure such as a heat map for example, but those results should have information about peptide count, coverage, etc in a sup data and all the full MS data sets available in a protein repository database.
Author Response
Reviewer 1
I have read the manuscript entitled: “A synthetic cytokinin improves photosynthesis in rice under drought stress by modulating the expression of proteins related to stomatal conductance, chlorophyll contents, and Rubisco activity” by Gujjar et al. In this research article the authors present the effect of cytokinin- like plant growth regulator on rice during water stress condition. Physiological parameters are analysed and protein abundances using MS approaches determined for some targeted proteins. Overall, the study is interesting, but the methods and presentation of the results should be improved for better clarity.
- To allow for the clear understanding and reliability of the MS results the LCMS data should be presented as sup data and the full MS data fully accessible in MS repository databases.
Response: Sir, based on your constructive suggestion, the mass spectrometry proteomics data have been deposited to the ProteomeXchange Consortium via the PRIDE [1] partner repository with the dataset identifier PXD021005, and the same has line been inducted in Materials and Methods.
- There is a need for a figure that summarizes the results obtained, ie physiological and protein data.
Response: Sir, based on your constructive suggestion, a figure summarizing the results of physiological and protein data, has been included in the conclusion section of the manuscript (Fig. 11).
- The wording “expression” for proteins is not adequate, ‘abundance’ is a better term and should be corrected accordingly in the manuscript.
Response: Sir, based on your constructive suggestion, we have replaced the term “expression” with “abundance” in the manuscript.
Methods should be more precise.
Line 92, Please add duration and if any washes done
Response: Sir, based on your constructive suggestion, we have added the duration of 30 min. Only one wash was given in the treatment.
Line 92. Was the paper left on the bench or in a container, with light or not, etc
Response: Sir, based on your constructive suggestion, we have modified the sentence as ‘Seeds were disinfected by sodium hypochlorite (Chlorox® 10% v/v) for 30 min. in 100 ml flask and were subsequently germinated in a container on moistened filter paper (Whatman® no. 1) for 14 days in light at room temperature.”
Line 93. What is an “experimental block”?, ie length, depth, etc. what is the number of plants per block, ie density.
Response: Sir, based on your constructive suggestion, we have added the details as “Thirty Seedlings (5 cm long, with true leaves) were transplanted with the spacing of 5 x 5 inch into the experimental blocks (length x width x height = 3 meter x 2 meter x 30 cm) filled with sand : soil (2:1) at the greenhouse, Salaya Campus, Mahidol University.”
Line 102. Any tween is the control, if yes please add.
Response: Sir, based on your constructive suggestion, we have included the following in the manuscript- “Control plants were sprayed with sterile water, added with 0.1% Tween 20® solution at the same time.”
Line 121. What is small pieces?
Response: Sir, based on your constructive suggestion, we have removed the ‘small’ word and modified the sentence as- “About 0.1 g of fresh flag leaf sample was cut into pieces and homogenized with 5 ml of 80 % acetone and were kept at 4 °C for 48 h.”
Line 130. How many plants per replicate?
Response: Sir, based on your constructive suggestion, we have modified the sentence as “Fresh flag leaf samples were collected in triplicates (biological) from all the treatment combinations of drought and CPPU for differential proteomic analysis.”
Line 135. Centrifugation steps at room temp?
Response: Sir, based on your constructive suggestion, we have modified the sentence, as “Samples were vortexed, incubated at -20 °C for 1 h, and centrifuged at 12,000 rpm for 5 min at 4°C.”
Line 137. How long the drying?
Response: Sir, based on your constructive suggestion, we have added the drying time as follows “The precipitates were dried in the oven at 55 °C for 30 min.”
Line 137. What was the volume/dried mass?
Response: Sir, as per the modified version of protein extraction protocol described by Shen et al. (2008), we did not weigh the dried samples.
Figures with vertical bars have asterisks to indicate the significance level but already indicated in the legend and is a bit awkward in the figure itself, I would remove it.
Response: Sir, based on your constructive suggestion, we have removed the asterisks and modified the figure 1, 6, 8 and 9.
Figures with protein abundance: Why there is no standard deviation presented while the methods indicate 3 replicates? The manuscript would benefit to have all those results presented in a single figure such as a heat map for example, but those results should have information about peptide count, coverage, etc in a sup data and all the full MS data sets available in a protein repository database.
Response: Sir, based on your constructive suggestion, we have modified the “materials and methods” as “the biological triplicates were merged due to large number of samples (line 144-145) and divided into technical triplicates for LC-MSMS analysis. The highest log2 abundance intensity value among the three technical replicates was used as representative value of that treatment (Line 172-173). The mass spectrometry proteomics data have been deposited to the ProteomeXchange Consortium via the PRIDE [1] partner repository with the dataset identifier PXD021005, and the same has line been inducted in ‘Materials and Methods’.

Reviewer 2 Report
In this manuscript, the authors investigated roles of synthetic cytokinin CPPU in regulating stomatal conductance, accumulation of photosynthetic pigments, photosynthesis rate and related protein levels under drought stress. This manuscript suggested positive roles of CPPU in alleviating drought stress and maintaining photosynthesis in drought susceptible cultivar. The manuscript conveys interesting observations and might contribute to the enhancement of agriculture confronting nature disasters. However, the data for levels of different proteins was presented using highest log2 fold change values of biological replicates, which is not appropriate and reduces the scientific soundness.
- Please include all values of biological replicates and use the mean value and standard deviation/standard error, or use box and whisker plot, to present the changes of protein levels in Figures and Table 1. To compare the significance of the changes of protein levels, statistical analysis must be carried out.
- Please make sure the conclusions or statements were consistent with the data presented.
For example:
Line 180-181 “the stomatal conductance of CPPU treated plants was significantly higher than that of untreated plants”: at tillering stage, Day 14, stomatal conductance of DR-CPPU plants was statistically similar to DR plants.
Line 205-206 “the CPPU treated plants, under drought stress, retained the normal expression levels of these proteins, similar to well-watered plants”: at least in tillering stage, ZEP levels in DR-CPPU plants looked more similar to the level in DR plants, and higher than that in WW plants.
Line 330-332 “However, in a sole incident during the grain-filling stage, drought-stressed plants had lower contents of Chl a, compared to well-watered plants on day 14 of drought stress”: during grain-filling stage, at Day 14, Chl a level in DR plants was not significantly different from that in WW plants. Moreover, Chl a level in DR-CPPU plants was significantly reduced compared to the level in WW plants.
Line 332-333 “Drought stress perpetually reduced the Chl b contents with an exception during the grain-filling stage on day 14 of drought stress”: No exception.
Line 334-335 “Interestingly, Chl b contents of CPPU treated plants under drought stress were almost comparable to those of well-watered plants”: only comparable at Day 7 tillering stage, Chl b was higher in DR-CPPU plants than that in WW plants at Day 14 in tillering stage, and at Day 7 and Day 14 in grain-filling stage.
Line 402-403 “exposure of drought alone, did not affect the expression of CCD8B protein significantly”: remarkable reduction could be found.
Line 403-406 “The results clearly corroborate with the concentration of carotenoids in CPPU treated plant under drought stress”: Compared between Figure 8A, B and C, several pairs of CCD8B protein level and corresponding carotenoid level were not co-related.
- Since the manuscript focused only on CPPU treatment, statements, such as “Plants were tested for the effect of cytokinin on the parameters influencing the process of photosynthesis” (L17-18), should be more accurate (“the effect of synthetic cytokinin”). Please go through the whole manuscript and make changes accordingly.
- Please indicate the numbers of biological replicates in all experiments either in Materials and Methods or figure legends.
- For using protein names (full name Vs abbreviation) in the main text and figures, what was the authors’ rationality? Please make changes so that the rationality is consistent throughout the manuscript.
Author Response
Reviewer 2
In this manuscript, the authors investigated roles of synthetic cytokinin CPPU in regulating stomatal conductance, accumulation of photosynthetic pigments, photosynthesis rate and related protein levels under drought stress. This manuscript suggested positive roles of CPPU in alleviating drought stress and maintaining photosynthesis in drought susceptible cultivar. The manuscript conveys interesting observations and might contribute to the enhancement of agriculture confronting nature disasters. However, the data for levels of different proteins was presented using highest log2 fold change values of biological replicates, which is not appropriate and reduces the scientific soundness.
- Please include all values of biological replicates and use the mean value and standard deviation/standard error, or use box and whisker plot, to present the changes of protein levels in Figures and Table 1. To compare the significance of the changes of protein levels, statistical analysis must be carried out.
Response: Sir, based on your constructive suggestion, we have modified the “materials and methods” as “the biological triplicates were merged due to large number of samples (line 144-145) and divided into technical triplicates for LC-MSMS analysis. The highest log2 abundance intensity value among the three technical replicates was used as representative value of that treatment (Line 172-173). The mass spectrometry proteomics data have been deposited to the ProteomeXchange Consortium via the PRIDE [1] partner repository with the dataset identifier PXD021005, and the same has line been inducted in ‘Materials and Methods’.
- Please make sure the conclusions or statements were consistent with the data presented.
For example:
Line 180-181 “the stomatal conductance of CPPU treated plants was significantly higher than that of untreated plants”: at tillering stage, Day 14, stomatal conductance of DR-CPPU plants was statistically similar to DR plants.
Response: Sir, based on your constructive suggestion, we have modified the sentence as “the stomatal conductance of CPPU treated plants was significantly higher than that of untreated plants, with an exception on day 14 at tillering stage.”
Line 205-206 “the CPPU treated plants, under drought stress, retained the normal expression levels of these proteins, similar to well-watered plants”: at least in tillering stage, ZEP levels in DR-CPPU plants looked more similar to the level in DR plants, and higher than that in WW plants.
Response: Sir, based on your constructive suggestion, we have modified the sentence as “In our study, both the proteins involved in ABA biosynthesis, ZEP and NCED, were more abundant in drought stressed plants at both day 7 and 14 of drought stress (Fig. 2). In contrast, the CPPU treated plants, under drought stress, retained the normal levels of these proteins, similar to well-watered plants. However, the abundance of ZEP in synthetic cytokinin treated plants looked similar to the levels in drought stressed plants at tillering stage.”
Line 330-332 “However, in a sole incident during the grain-filling stage, drought-stressed plants had lower contents of Chl a, compared to well-watered plants on day 14 of drought stress”: during grain-filling stage, at Day 14, Chl a level in DR plants was not significantly different from that in WW plants. Moreover, Chl a level in DR-CPPU plants was significantly reduced compared to the level in WW plants.
Response: Sir, based on your constructive suggestion, we have corrected the sentence as However, in a sole incident during the grain-filling stage, the well-watered plants had higher contents of Chl a, compared to drought stressed plants on day 14 of drought stress.
Line 332-333 “Drought stress perpetually reduced the Chl b contents with an exception during the grain-filling stage on day 14 of drought stress”: No exception.
Response: Sir, based on your constructive suggestion, we have corrected the sentence as “Drought stress perpetually reduced the Chl b contents in the leaves at both the growth stages of rice.”
Line 334-335 “Interestingly, Chl b contents of CPPU treated plants under drought stress were almost comparable to those of well-watered plants”: only comparable at Day 7 tillering stage, Chl b was higher in DR-CPPU plants than that in WW plants at Day 14 in tillering stage, and at Day 7 and Day 14 in grain-filling stage.
Response: Sir, based on your constructive suggestion, we have corrected the sentence as ‘Interestingly, Chl b content of CPPU treated plants, under drought stress, was higher than that in well-watered plants at Day 14 in tillering stage, and at Day 7 and Day 14 in grain-filling stage.
Line 402-403 “exposure of drought alone, did not affect the expression of CCD8B protein significantly”: remarkable reduction could be found.
Response: Sir, based on your constructive suggestion, we have removed this line/sentence from the manuscript.
Line 403-406 “The results clearly corroborate with the concentration of carotenoids in CPPU treated plant under drought stress”: Compared between Figure 8A, B and C, several pairs of CCD8B protein level and corresponding carotenoid level were not co-related.
Response: Sir, based on your constructive suggestion, we have corrected it as “The results largely corroborated with the concentration of carotenoids in synthetic cytokinin treated plant at grain-filling under drought stress.”
- Since the manuscript focused only on CPPU treatment, statements, such as “Plants were tested for the effect of cytokinin on the parameters influencing the process of photosynthesis” (L17-18), should be more accurate (“the effect of synthetic cytokinin”). Please go through the whole manuscript and make changes accordingly.
Response: Sir, based on your constructive suggestion, we have incorporated the term “synthetic cytokinin” throughout the manuscript at the places where we mentioned “cytokinin”
- Please indicate the numbers of biological replicates in all experiments either in Materials and Methods or figure legends.
Response: Sir, based on your constructive suggestion, we have indicated the number of biological replicates (3) in the Materials and Methods of the manuscript.
- For using protein names (full name Vs abbreviation) in the main text and figures, what was the authors’ rationality? Please make changes so that the rationality is consistent throughout the manuscript.
Response: Sir, based on your constructive suggestion, we have changed the figures and figure legends systematically. Abbreviations of proteins have been used in all the figures and expanded in the figure legends.
Round 2
Reviewer 1 Report
The authors have improved their manuscript by addressing most of my suggestions, but some still need to be addressed before final acceptation for publication.
While the authors replace “expression” by “abundance” in the text, the figures should be modified as well.
The authors have made publicly available their data but there is also a need for a supplemental data with the all the presented proteins that show protein ID (ie Uniprot), BLAST results, % coverage, number of peptides identified and other relevant information.
Author Response
The authors have improved their manuscript by addressing most of my suggestions, but some still need to be addressed before final acceptation for publication.
- While the authors replace “expression” by “abundance” in the text, the figures should be modified as well.
Response: Sir, based on your constructive suggestion, we have modified the figures by replacing the term “expression” by “abundance”
- The authors have made publicly available their data but there is also a need for a supplemental data with the all the presented proteins that show protein ID (ie Uniprot), BLAST results, % coverage, number of peptides identified and other relevant information.
Response: Sir, based on your constructive suggestion, we have attached a supplementary EXCEL file containing the required information about the proteins mentioned in the manuscript.

Reviewer 2 Report
The manuscript has been largely improved after the revision. However, some questions were not accurately answered and there are also other aspects that need to be modified, according to reviewer’s previous questions and/or the authors’ responses.
- According to the authors: “the biological triplicates were merged due to large number of samples (line 144-145) and divided into technical triplicates for LC-MSMS analysis. The highest log2 abundance intensity value among the three technical replicates was used as representative value of that treatment (Line 172-173)”.
In this case, since statistical analysis for each comparison is impossible, please indicate the changes with numbers between each comparisons in the main text and also label the numbers of protein abundance in the figures. Otherwise, all statements, for comparisons of protein abundance made via the length of bars and eye only, were not apparent.
For example:
Line 213-214: “In our study, both the proteins involved in ABA biosynthesis, ZEP and NCED, were overexpressed more abundant in drought stressed plants at both day 7 and 14 of drought stress”, should be “In our study, both the proteins involved in ABA biosynthesis, ZEP and NCED, were overexpressed more abundant (?? fold higher than well-watered) in drought stressed plants at both day 7 and 14 of drought stress”.
- Again, authors need to indicate the number of biological replicates for all experiments, including stomatal conductance, net photosynthetic rate, and spectrophotometric analysis of photosynthetic pigments.
- Line 342-343 “the well-watered plants had higher contents of Chl a, compared to drought stressed plants on day 14 of drought stress”. Again, there were no significant difference according to statistical analysis (“a” Vs “ab”).
Author Response
The manuscript has been largely improved after the revision. However, some questions were not accurately answered and there are also other aspects that need to be modified, according to reviewer’s previous questions and/or the authors’ responses.
- According to the authors: “the biological triplicates were merged due to large number of samples (line 144-145) and divided into technical triplicates for LC-MSMS analysis. The highest log2 abundance intensity value among the three technical replicates was used as representative value of that treatment (Line 172-173)”.
In this case, since statistical analysis for each comparison is impossible, please indicate the changes with numbers between each comparisons in the main text and also label the numbers of protein abundance in the figures. Otherwise, all statements, for comparisons of protein abundance made via the length of bars and eye only, were not apparent.
For example:
Line 213-214: “In our study, both the proteins involved in ABA biosynthesis, ZEP and NCED, were overexpressed more abundant in drought stressed plants at both day 7 and 14 of drought stress”, should be “In our study, both the proteins involved in ABA biosynthesis, ZEP and NCED, were overexpressed more abundant (?? fold higher than well-watered) in drought stressed plants at both day 7 and 14 of drought stress”.
Response: Sir, based on your constructive suggestion, we have modified/revised the figures by adding the values of Log2 abundance intensities of the proteins.
2. Again, authors need to indicate the number of biological replicates for all experiments, including stomatal conductance, net photosynthetic rate, and spectrophotometric analysis of photosynthetic pigments.
Response: Sir, based on your useful suggestion, we have included the biological replicates (3) for stomatal conductance, net photosynthetic rate, and spectrophotometric analysis of photosynthetic pigments.
3. Line 342-343 “the well-watered plants had higher contents of Chl a, compared to drought stressed plants on day 14 of drought stress”. Again, there were no significant difference according to statistical analysis (“a” Vs “ab”).
Response: Sir, based on your constructive suggestion, we have removed the wrongly interpreted sentence.
